# UniHand: A Unified Model for Diverse Controlled 4D Hand Motion Modeling

**Zhihao Sun**[1,2]**, Tong Wu**[3]**, Ruirui Tu**[1,2]**, Daoguo Dong**[1,2†]**, Zuxuan Wu**[1,2†]

[1]Institute of Trustworthy Embodied AI, Fudan University
[2]Shanghai Key Laboratory of Multimodal Embodied AI
[3]Stanford University
[†]Correspondence Authors

## Abstract

Hand motion plays a central role in human interaction, yet modeling realistic 4D hand motion (*i.e.*, 3D hand pose sequences over time) remains challenging. Research in this area is typically divided into two tasks: (1) Estimation approaches reconstruct precise motion from visual observations, but often fail under hand occlusion or absence; (2) Generation approaches focus on synthesizing hand poses by exploiting generative priors under multi-modal structured inputs and infilling motion from incomplete sequences. However, this separation not only limits the effective use of heterogeneous condition signals that frequently arise in practice, but also prevents knowledge transfer between the two tasks. We present **UniHand**, a unified diffusion-based framework that formulates both estimation and generation as conditional motion synthesis. UniHand integrates heterogeneous inputs by embedding structured signals into a shared latent space through a joint variational autoencoder, which aligns conditions such as MANO parameters and 2D skeletons. Visual observations are encoded with a frozen vision backbone, while a dedicated hand perceptron extracts hand-specific cues directly from image features, removing the need for complex detection and cropping pipelines. A latent diffusion model then synthesizes consistent motion sequences from these diverse conditions. Extensive experiments across multiple benchmarks demonstrate that UniHand delivers robust and accurate hand motion modeling, maintaining performance under severe occlusions and temporally incomplete inputs.

## 1 Introduction

The human hand plays a central role in our interactions with the world. It not only allows us to manipulate tools with dexterity but also to communicate through gestures. Given this importance, modeling realistic 4D hand motion (*i.e.*, 3D hand pose sequences over time) has emerged as an active research problem in computer vision and graphics. Progress in this field is crucial for applications such as virtual reality (VR), digital avatars, and robotics (Qi et al., 2024; Xie et al., 2025; Jiang et al., 2025).

Existing research in 4D hand modeling is predominantly divided into two distinct tasks, each typically addressed by specialized models. Estimation approaches aim to reconstruct precise motion directly from visual observations, such as monocular or multi-view videos. These methods, however, often struggle with hand occlusions (Duran et al., 2024), temporally incomplete frames (Pavlakos et al., 2024; Dong et al., 2024), and tasks requiring flexible editing. Generation approaches, on the other hand, focus on synthesizing hand poses by exploiting generative priors under multi-modal structured inputs, such as 2D and 3D skeletons (Wan et al., 2017; Yang et al., 2019; Li et al., 2024), and infilling motions from incomplete sequences (Zhang et al., 2025; Yu et al., 2025).

This separation between estimation and generation not only restricts the effective use of heterogeneous condition signals that commonly arise in real-world scenarios, but also prevents the transfer of knowledge and motion priors across the two tasks. When accurate reconstruction is required, rich visual observations, such as images or videos, are indispensable. In contrast, for motion synthesis or editing, structured conditions such as 2D skeleton keypoints and MANO parameters are often more

suitable due to their ease of manipulation. In practice, visual inputs may be affected by hand occlusions or absence, while other condition signals can exhibit temporal discontinuities. These diverse and potentially incomplete conditions underscore the need for a unified framework that can flexibly integrate heterogeneous conditions and information.

Recent research has highlighted the potential synergy between estimation and generation. Some studies adopt multi-stage frameworks that exploit generative priors to refine or complete the hand pose sequences detected by estimation methods (Zhang et al., 2025; Yu et al., 2025). Other works explore unified generative approaches that support multiple modalities of input, thereby bridging the two tasks within a single formulation (Li et al., 2024). Building on these insights, we further extend this direction by exploring multimodal alignment and flexible condition integration, and introduce **UniHand**, a unified diffusion-based framework for 4D hand motion modeling under heterogeneous conditions. **For structured signals** such as MANO parameters and 2D skeleton keypoints, UniHand employs a joint variational autoencoder to align multiple encoders within a shared latent space, enabling all structured signals to be fused during the diffusion process. **For visual observations**, which are common and information-rich, particularly in estimation scenarios, UniHand uses a frozen vision backbone to extract features from full-size frames and a hand perceptron module to attend to hand-relevant tokens. A latent diffusion model then integrates multiple conditions to generate the final motion sequence. Motion is generated in a canonical camera space defined by the first frame, ensuring consistency under both static and dynamic cameras without relying on extrinsic calibration. By integrating diverse structured and visual conditions, UniHand unifies accurate estimation and flexible generation within a single framework. Our contributions can be summarized as follows:

- We propose UniHand, the first unified model that formulates both 4D hand motion estimation and generation as conditional motion synthesis. Our diffusion-based model flexibly integrates heterogeneous conditions.

- We design a joint variational autoencoder that aligns structured signals into a shared latent space, and introduce a hand perceptron module that directly attends to hand-related features from dense tokens extracted from full-size frames.

- We conduct extensive experiments on multiple benchmarks and demonstrate that UniHand achieves robust and accurate motion generation, particularly under challenging scenarios such as severe hand occlusions and temporally incomplete signals.

## 2 RELATED WORKS

### 2.1 HAND MOTION ESTIMATION

We first review research on hand pose estimation, where methods take visual observations as input to reconstruct hand pose or motion. Early works relied on depth cameras to reconstruct 3D hands (Ge et al., 2016; Oikonomidis et al., 2011). With the introduction of the MANO parametric hand model (Romero et al., 2017b), Boukhayma et al. (2019) proposed the first learning-based approach that directly regresses MANO parameters from RGB inputs, inspiring a line of follow-up studies (Baek et al., 2019; Zhang et al., 2019). Other works adopt a non-parametric strategy and directly predict the 3D mesh vertices of the MANO model (Kulon et al., 2019; Ge et al., 2019; Choi et al., 2020; Lin et al., 2021b). Recent studies have emphasized the importance of scaling both data and model capacity. HaMeR (Pavlakos et al., 2024) investigates this direction by combining large-scale training data with large Vision Transformers (ViT), while WiLoR (Potamias et al., 2025) introduces a data-driven pipeline and refinement strategy for efficient multi-hand reconstruction.

While most approaches focus on image-based estimation, they can also be directly applied to videos. However, this often ignores the temporal information contained in videos and struggles with challenges such as occlusions and fast motion. Deformer (Fu et al., 2023) implicitly reasons about the relationship between hand parts within the same image and across timesteps. HMP (Duran et al., 2024) exploits motion priors to enable video-based hand motion estimation through latent optimization. HaWoR (Zhang et al., 2025) reconstructs hand motion by decoupling hand pose reconstruction in camera space from camera trajectory estimation in the world frame. Dyn-HaMR (Yu et al., 2025) extends this idea with a multi-stage, multi-objective optimization pipeline that relies on external hand pose tracking and SLAM methods to model interacting hands under dynamic cameras. However, existing methods generally rely on multi-stage detection-based pipelines and cannot flexibly

incorporate diverse types of conditions. In this work, we instead view hand pose estimation as a special case of conditional motion synthesis, which enables a unified hand motion generation.

## 2.2 Hand Motion Generation

Human motion generation has been widely studied under diverse condition signals, including text (Tevet et al., 2023b; Jin et al., 2023), actions (Guo et al., 2020), speech (Alexanderson et al., 2023), music (Tseng et al., 2023), and scene (Hassan et al., 2021; Yi et al., 2024). In contrast, hand motion has not typically been conditioned on such a broad range of modalities. Most existing approaches focus on hand-object interactions (HOI), where object geometry serves as the primary prior for synthesizing plausible grasps and interaction sequences. For example, GraspDiff (Zuo et al., 2024) leverages diffusion models to directly generate grasps conditioned on 3D object models, while MGD (Cao et al., 2024) learns a joint prior across heterogeneous hand–object datasets for improved generalization. Sequential extension such as Text2HOI (Cha et al., 2024) incorporates text guidance by decomposing the task into contact and motion generation. Despite these advances, the reliance on object-specific priors and task-specific pipelines limits their applicability to broader hand motion modeling.

A more general direction explores probabilistic models to learn the distribution of feasible hand poses and motions. Unconditional priors aim to capture the distribution $p(x)$ of plausible hand poses without external inputs. Early approaches relied on biomechanical constraints, manually defining joint degrees of freedom and rotation ranges (Yang et al., 2021; Spurr et al., 2020). Later studies adopted data-driven strategies, such as applying principal component analysis (PCA) to MANO parameters (Romero et al., 2017a) or training variational autoencoders that map hand poses into Gaussian latent spaces (Zuo et al., 2023). Conditional priors instead model the distribution $p(x|c)$ under external conditions such as RGB images, depth maps, or 2D skeletons. Typical designs employ VAEs constructed in different domains and align their latent spaces to learn feasible hand configurations across modalities (Wan et al., 2017; Yang et al., 2019). More advanced formulations leverage score-based models to estimate the pose distribution (Ci et al., 2023). However, these approaches remain restricted to single-condition settings and struggle with temporally incomplete condition signals. In contrast, our framework employs a diffusion-based generative model that unifies diverse signals in a shared latent space and leverages vision inputs to capture hand-related features, enabling accurate 4D hand motion modeling under multimodal conditions.

## 3 Unified Model for Hand Motion Modeling

### 3.1 Preliminaries

**Problem Definition.** UniHand formulates hand motion estimation and generation within a unified framework of conditional hand motion generation. Specifically, it synthesizes a hand motion sequence $x = \{x^i\}_{i=1}^N$ of length $N$ based on a set of condition signals $C$ and a set of corresponding condition masks $M$. The condition set $C$ includes: video frames $c_{\text{vision}} \in \mathbb{R}^{N \times H \times W \times 3}$, 2D skeleton keypoints $c_{\text{2D}} \in \mathbb{R}^{N \times 21 \times 2}$, 3D skeleton keypoints $c_{\text{3D}} \in \mathbb{R}^{N \times 21 \times 3}$, and optionally hand pose parameters $\widetilde{x}$. Each condition $c \in C$ is paired with a binary mask $m \in \mathbb{R}^N$, where $m^i = 1$ if the condition signal is available at frame $i$, and $m^i = 0$ otherwise. This formulation allows the model to flexibly handle varying combinations of condition signals across frames.

**Hand Pose and Other Conditions Representation.** The 3D hand representation $x^i$ is parameterized by the MANO model (Romero et al., 2017b), and includes hand pose $\Theta^i \in \mathbb{R}^{15 \times 3}$, shape $\beta^i \in \mathbb{R}^{10}$, along with global orientation $\Phi^i \in \mathbb{R}^3$ and root translation $\Gamma^i \in \mathbb{R}^3$. For 3D hand estimation, hand poses are typically represented in the camera coordinate space to ensure better alignment with image features. However, for videos with dynamic camera perspectives, the hand motion sequence $x$ becomes discontinuous due to changing coordinate systems. While the world coordinate system can alleviate this issue, it does not facilitate alignment with visual observations. To address this, we introduce a canonical coordinate system, defined as the camera space of the first frame. This decouples the hand motion from the dynamic camera, providing a consistent representation across the entire sequence, while remaining applicable to both static and dynamic camera scenar-

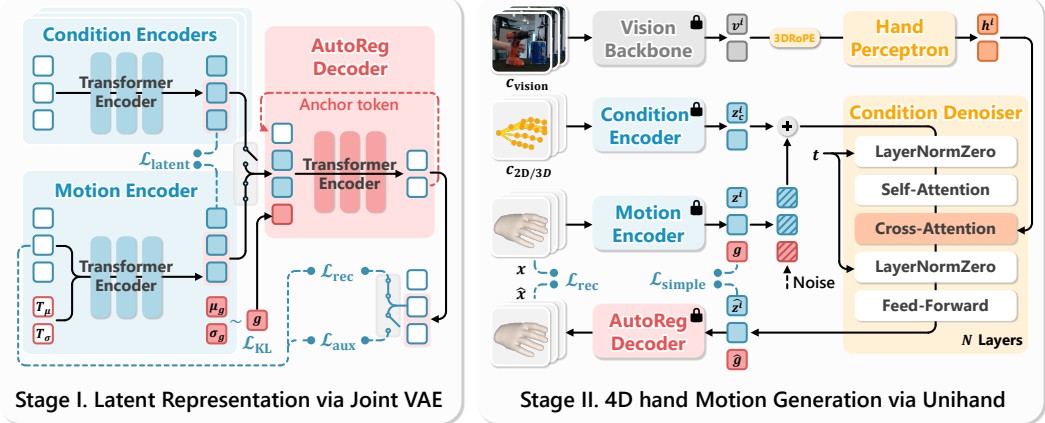

Figure 1: **Overview of the UniHand framework.** (I) The Joint VAE aligns motion and condition encoders within a shared latent space. An autoregressive decoder iteratively reconstructs motion to preserve temporal consistency. (II) The latent diffusion model is trained on this latent space, where multimodal conditions are fused, and hand-relevant vision tokens are integrated into the denoiser.

ios. Consequently, the 3D keypoint conditions are transformed into the canonical space to ensure consistency. More details are provided in Appendix A.1.

**Overview.** We propose a unified framework for conditional hand motion generation, which consists of a joint variational autoencoder (Joint VAE) and a latent diffusion model. The Joint VAE (Section 3.2) comprises multiple encoders for different modalities and a shared decoder, which together tokenize motion sequences and condition signals into a shared latent representation. The latent diffusion model (Section 3.3) is defined on this latent space, where it integrates hand-relevant vision features and multiple conditions. The framework is illustrated in Figure 1.

## 3.2 JOINT LATENT REPRESENTATION

Variational Autoencoders (VAEs) (Kingma & Welling, 2014) compress raw data into a latent space and have proven effective in learning compact yet expressive representations. Encoding motion in this latent space mitigates the temporal inconsistencies that often arise when training diffusion models directly on raw motion sequences (Chen et al., 2023; Zhao et al., 2025). We propose a Joint VAE that encodes both motion sequences and diverse condition signals into a shared latent space. This alignment between MANO-based motion, 2D skeleton keypoints, and 3D skeleton keypoints encourages the latent representation to capture motion semantics that generalize across modalities. The shared space further facilitates flexible condition fusion during controllable generation.

As shown in Algorithm 1, we design a joint encoder architecture that incorporates both a motion encoder and multiple condition encoders. The motion encoder $\mathcal{E}_m$ encodes the sequence $x = \{x^i\}_{i=1}^N$ into a set of latent motion tokens $z = \{z^i\}_{i=1}^N$, where each token $z^i$ represents the hand pose of a single frame in a $d$-dimensional latent space. In addition, a global motion token $g \in \mathbb{R}^d$ is introduced to capture sequence-level information. We introduce learnable distribution tokens $T_\mu, T_\sigma$, and the encoder predicts Gaussian parameters $(\mu_g, \sigma_g)$ from which $g$ is sampled. This latent variable is regularized via a KL divergence loss. Similarly, each condition encoder $\mathcal{E}_c$ tokenize a condition signal $c \in C$ into a sequence of condition latent tokens $z_c = \mathcal{E}_c(c) \in \mathbb{R}^{N \times d}$, which are aligned in the shared latent space and can be fused during generation. The decoder $\mathcal{D}$ reconstructs the motion sequence $x$ in an autoregressive manner. At each autoregression step, it predicts a motion segment $\hat{x}^{i:i+n}$ conditioned on the latent tokens $z^{i:i+n}$, the global token $g$, and an anchor token $a^i$ representing the initial state of the segment. The global token provides high-level structural context, while the frame-wise latent tokens preserve fine-grained motion details and condition alignment. The training objective is provided in Appendix A.2.

---

**Algorithm 1** Latent representation with Joint Variational Autoencoder

---

**Input:** hand motion $x = \{x^i\}_{i=1}^N$, structured conditions $c = \{c^i\}_{i=1}^N$, motion encoder $\mathcal{E}_m$, learnable distribution tokens $T_\mu$ and $T_\sigma$, condition encoders $\mathcal{E}_c$, and autoregressive decoder $\mathcal{D}$.

**Output:** motion latent tokens $z = \{z^i\}_{i=1}^N$, motion global token $g$, condition latent tokens $z_c = \{z_c^i\}_{i=1}^N$, and reconstructed hand motion $\hat{x}$.

1:  $(z, \mu_g, \sigma_g) \leftarrow \mathcal{E}_m(x, T_\mu, T_\sigma)$                 ▷ encode hand motion to latent representation
2:  $g \sim \mathcal{N}(\mu_g, \sigma_g)$                                      ▷ sample motion global token
3:  **for** $c$ in $C$ **do**
4:      $z_c \leftarrow \mathcal{E}_c(c)$               ▷ encode each structured condition to latent representation
5:  **end for**
6:  $\hat{x} \leftarrow \emptyset$, $a^1 \leftarrow \text{Linear}(x^1)$              ▷ initialize reconstructed motion and anchor token
7:  **for** $z$ in $\{z, z_c\}$ **do**
8:      **for** $i = 1$ to $N$ by step size $n$ **do**                    ▷ autoregressive rollouts
9:          $\hat{x}^{i:i+n} \leftarrow \mathcal{D}(a^i, z^{i:i+n}, g)$    ▷ autoregressive decoding with anchor and global token
10:         $\hat{x} \leftarrow \text{CONCAT}(\hat{x}, \hat{x}^{i:i+n})$
11:         $a^{i+n} \leftarrow \text{Linear}(\hat{x}^{i+n-1})$                      ▷ update anchor token
12:     **end for**
13: **end for**
14: **return** $z, g, z_c, \hat{x}$

---

## 3.3 DIFFUSION-BASED MOTION GENERATION

We perform diffusion-based generation in the latent space learned by the Joint VAE. Diffusion models (Ho et al., 2020) define a stochastic process that iteratively adds Gaussian noise to a clean latent representation until it becomes pure Gaussian noise, and then learns to reverse the process for generation. Given a hand motion sequence $x$ and its latent representation $z_0 \in \mathbb{R}^{N \times d}$ obtained by the encoder $\mathcal{E}$, The forward process progressively transforms $z_0$ into Gaussian noise $z_T \sim \mathcal{N}(0, I)$ through a Markov chain: $q(z_t \mid z_{t-1}) = \mathcal{N}(\sqrt{1 - \beta_t}\, z_{t-1}, \beta_t I)$, where $\{\beta_t\}$ is a predefined noise schedule. The denoiser model $\mathcal{G}_\theta$ learns the reverse process, which aims to transform noise back into clean motion latents conditioned on $C$: $p_\theta(z_{t-1} \mid z_t, C) = \mathcal{N}(\mu_\theta(z_t, t, C), \Sigma_t I)$, where $C$ denotes the available conditions, such as vision frames and 2D skeleton keypoints, and $\Sigma_t$ is determined by the noise schedule. Following prior work in human motion generation (Shafir et al., 2023; Tevet et al., 2023a; Zhao et al., 2025), which show that predicting the clean sample yields more temporally coherent motions than predicting noise, we design the denoiser $\mathcal{G}_\theta$ to predict the clean latent $\hat{z}_0 = \mathcal{G}_\theta(z_t, t, C)$. The predicted $\hat{z}_0$ is then used to compute the mean of the reverse distribution:

$$\mu_t = \frac{\sqrt{\bar{\alpha}_{t-1}}\beta_t}{1 - \bar{\alpha}_t}\, \hat{z}_0 + \frac{\sqrt{\alpha_t}(1 - \bar{\alpha}_{t-1})}{1 - \bar{\alpha}_t}\, z_t, \tag{1}$$

with $\alpha_t = 1 - \beta_t$ and $\bar{\alpha}_t = \prod_{i=1}^t \alpha_i$. Following Yang et al. (2024), we incorporate the diffusion timestep $t$ into the modulation module of an adaptive LayerNorm.

**Attending to Hand-relevant Vision Tokens.** Visual observations, such as images and videos, are the most common inputs in hand pose estimation and provide the richest information among all modalities. They capture not only hand pose but also contextual cues from the surrounding environment and interacting objects. However, existing approaches often crop around the hand region, which sacrifices contextual information and, in the case of video, disrupts temporal consistency since the camera coordinates of the cropped regions differ across time. We instead leverage a pretrained vision backbone $\mathcal{E}_{\text{vision}}$ to process a full image or video frame $c_{\text{vision}}^i$ and project it into dense tokens $v^i \in \mathbb{R}^{h \times w \times d}$. To extract hand-relevant information from these dense features, we introduce a hand perceptron module that selectively attends to hand-related vision tokens while retaining contextual cues from the environment and interacting objects. Specifically, we employ a set of trainable hand tokens $H = \{H^i\}_1^N$, along with an initialization hand pose token $a^1$, as queries. The dense vision tokens $v$ serve as keys and values. We adopt Rotary Positional Encoding (RoPE) (Su et al., 2024) in 3D formation, following prior work (Kong et al., 2024; Yang et al., 2024), and compute the rotary frequency matrices separately for the temporal $N$, height $h$, and width $w$ dimensions of the vision

Table 1: Quantitative comparison of SoTA hand pose and motion modeling methods on the DexYCB test set in the camera coordinate space. Results are reported in terms of MPJPE (mm) and AUC, with statistics across different occlusion levels.

| Method | All | | Occlusion (25%–50%) | | Occlusion (50%–75%) | | Occlusion (75%–100%) | |
|---|---|---|---|---|---|---|---|---|
| | PA-MPJPE $\downarrow$ | AUC$_J\uparrow$ | PA-MPJPE $\downarrow$ | AUC$_J\uparrow$ | PA-MPJPE $\downarrow$ | AUC$_J\uparrow$ | PA-MPJPE $\downarrow$ | AUC$_J\uparrow$ |
| Spurr et al. (2020) | 6.83 | 0.864 | 7.22 | 0.856 | 8.00 | 0.840 | 10.65 | 0.788 |
| MeshGraphormer | 6.41 | 0.872 | 6.85 | 0.863 | 7.22 | 0.856 | 7.76 | 0.845 |
| SemiHandObj | 6.33 | 0.874 | 6.70 | 0.866 | 7.17 | 0.857 | 8.96 | 0.821 |
| HandOccNet | 5.80 | 0.884 | 6.22 | 0.876 | 6.43 | 0.872 | 7.37 | 0.853 |
| WiLoR | 5.01 | 0.900 | - | - | 5.42 | 0.892 | 5.68 | 0.887 |
| $S^2$HAND(V) | 7.27 | 0.855 | 7.74 | 0.845 | 7.71 | 0.846 | 7.87 | 0.843 |
| VIBE | 6.43 | 0.871 | 6.72 | 0.865 | 6.84 | 0.864 | 7.06 | 0.858 |
| TCMR | 6.28 | 0.875 | 6.56 | 0.869 | 6.58 | 0.868 | 6.95 | 0.861 |
| Deformer | 5.22 | 0.896 | 5.71 | 0.886 | 5.70 | 0.886 | 6.34 | 0.873 |
| HaWoR | 4.76 | 0.905 | - | - | 5.03 | 0.899 | 5.07 | 0.899 |
| **UniHand** | **4.08** | **0.918** | **4.22** | **0.913** | **4.25** | **0.912** | **4.26** | **0.912** |

tokens. The attention mechanism is then applied as:

$$\text{Attention}(\mathbf{Q}, \mathbf{K}, \mathbf{V}) = \text{Softmax}(\mathbf{Q}\mathbf{K}^T/\sqrt{d_k})\mathbf{V}, \tag{2}$$

$$\mathbf{Q} = \text{RoPE}(\text{LayerNorm}(W_{\mathbf{Q}}(a^1, H), P_{\text{1D}})),$$
$$\mathbf{K} = \text{RoPE}(\text{LayerNorm}(W_{\mathbf{K}}(v), P_{\text{3D}})), \tag{3}$$
$$\mathbf{V} = \text{LayerNorm}(W_{\mathbf{V}}(v)).$$

The trainable hand tokens aggregate vision information associated with the target hand in each frame, while the initialization pose token anchors the attention process to the correct hand instance when multiple hands are present, thereby ensuring a consistent one-to-one binding across the sequence. As a result, the hand perceptron produces a single hand token $h^i$ for each frame.

**Integrating Multiple Conditions.** Our framework supports multiple forms of conditions, which can be grouped into structured conditions and visual observations. The first group includes signals such as MANO parameters, 2D keypoints, and 3D keypoints. These representations are encoded into the shared latent space by the Joint VAE and can therefore be directly fused with the noisy motion latent during denoising. The second group consists of visual inputs, from which we extract one representative hand token per frame. Rather than being fused at the latent level, these tokens are incorporated into the denoising network through attention layers at every denoising step, allowing the model to attend to vision information throughout the generation process.

We adopt a two-stage training strategy, where the Joint VAE and the diffusion model are trained separately, with details provided in Appendix B.2. To further enhance generation quality and condition flexibility, we adopt classifier-free guidance (CFG) (Ho & Salimans, 2022) with trainable unconditional tokens. CFG is typically expressed as $\hat{\mathcal{G}}_\theta = \mathcal{G}\theta(z_t, t, c_\varnothing) + w\big(\mathcal{G}\theta(z_t, t, c_t) - \mathcal{G}\theta(z_t, t, c_\varnothing)\big)$, where $\mathcal{G}$ denotes the denoising network, $z_t$ is the noisy latent at timestep $t$, and $w$ is the CFG scale controlling the strength of condition. However, motion latents do not possess natural unconditional forms $c_\varnothing$. To address this, we introduce independent learnable unconditional tokens for motion and condition representations, which match the feature dimensions of $z$ and $z_c$, respectively. During training, a condition latent $z_c^t$ is randomly replaced with its unconditional form $z_{c_\varnothing}$ with a predefined probability $p$. This mechanism ensures that UniHand remains robust under diverse and potentially incomplete conditioning scenarios, while also allowing fine-grained adjustment of conditional influence during motion synthesis. Further details on training and inference are provided in the Appendix A.3.

## 4 EXPERIMENTS

### 4.1 EXPERIMENTAL SETUP

**Datasets.** To evaluate the performance of UniHand under egocentric views with dynamic cameras and to compare it with existing methods, we use the DexYCB dataset (Chao et al., 2021), which

Table 2: Quantitative comparison of baseline hand pose estimation methods on the HO3D dataset in the camera coordinate space. Results are reported in terms of MPJPE ($mm$), AUC scores, and F-scores.

| Method | PA-MPJPE ↓ | AUC$_J$ ↑ | F@5 ↑ | F@15 ↑ |
|---|---|---|---|---|
| HandOccNet | 9.1 | 0.819 | 0.564 | 0.963 |
| AMVUR | 8.3 | 0.835 | 0.608 | 0.965 |
| HaMeR | 7.7 | 0.846 | 0.635 | 0.980 |
| WiLoR | 7.5 | 0.851 | 0.646 | 0.983 |
| Deformer | 9.4 | - | 0.546 | 0.963 |
| **Ours** | **6.7** | **0.866** | **0.671** | **0.988** |

Table 3: Quantitative evaluation of SoTA methods on the HOT3D dataset in the world coordinate space. Results are reported in terms of MPJPE ($mm$) under different alignment strategies and acceleration error.

| Method | PA-MPJPE ↓ | G-MPJPE ↓ | GA-MPJPE ↓ | AccEr ↓ |
|---|---|---|---|---|
| HaMeR-SLAM | 9.20 | 161.31 | 43.85 | 15.53 |
| WiLoR-SLAM | 7.17 | 154.74 | 40.69 | 10.39 |
| HMP-SLAM | 10.68 | 128.56 | 38.25 | 5.41 |
| Dyn-HaMR | 8.92 | 59.04 | 23.57 | 5.16 |
| HaWoR | 5.47 | **47.35** | **18.14** | 5.88 |
| **Ours** | **4.76** | 63.97 | 25.24 | **4.93** |

contains multi-view videos with hand pose annotations in the camera coordinate system. The degree of occlusion can be computed, enabling analysis of pose estimation under different occlusion levels. We further report results on HO3D (Hampali et al., 2020) to assess the generalization ability of UniHand. Following Zhang et al. (2025); Yu et al. (2025), we also use HOT3D (Banerjee et al., 2025), which provides hand poses in the world coordinate system along with camera extrinsics, to evaluate estimation performance under egocentric views with dynamic cameras.

**Metrics.** We report Procrustes-Aligned Mean Per-Joint Position Error (PA-MPJPE) and the area under the curve of correctly localized keypoints (AUC$_J$) to evaluate hand pose in the camera coordinate space. Following Hampali et al. (2020), we also include the fraction of poses with less than $5mm$ and $15mm$ error (F@5, F@15) computed by the official evaluation scripts. In the world coordinate space, we report G-MPJPE and GA-MPJPE following Ye et al. (2023), where alignment with ground truth is performed using the first two frames or the entire motion. In addition, we compute the acceleration error (AccEr) to assess the temporal smoothness of the generated motion.

## 4.2 Hand Motion in Camera Coordinate Space

Hand pose estimation in the camera coordinate space provides the most direct way to evaluate the quality of motion generation conditioned on visual observations. Moreover, evaluation under challenging conditions such as occlusions and missing temporal frames is particularly important, as these phenomena frequently occur in real-world videos. Following prior work (Fu et al., 2023; Zhang et al., 2025), we evaluate our method on DexYCB, a dataset that provides frame-level occlusion-related annotations. We partition the test set into multiple occlusion levels. For our approach and other video-based methods, we use videos as input and then compute frame-level metrics, ensuring fair comparison with image-based methods.

As shown in Table 1, we compare UniHand against a wide range of image-based and video-based baselines across different occlusion categories. Image-based approaches include Mesh-Graphormer (Lin et al., 2021a), SemiHandObj (Liu et al., 2021), HandOccNet (Park et al., 2022), and WiLoR (Potamias et al., 2025), which process images independently and are typically sensitive to occlusion. In contrast, video-based methods such as $S^2$HAND(V) (Tu et al., 2023), VIBE (Kocabas et al., 2020), TCMR (Choi et al., 2021), Deformer (Fu et al., 2023), and HaWoR (Zhang et al., 2025) leverage temporal context for motion reasoning and are therefore less affected by occlusion. UniHand achieves a PA-MPJPE of 4.08 and an AUC of 0.918, outperforming all image-based and video-based baselines. Even under the most severe occlusion level, our method maintains superior performance with PA-MPJPE of 4.26 and AUC of 0.912. These results highlight not only the benefit of temporal modeling but also the advantages of our generative priors and the hand perceptron module in effectively exploiting visual input.

To further evaluate generalization, we evaluate our model on the HO3D dataset, which contains diverse object interaction scenarios and severe occlusions not present in the training data. As shown in Table 2, despite the domain shift, our model achieves competitive performance, demonstrating robustness to out-of-distribution inputs.

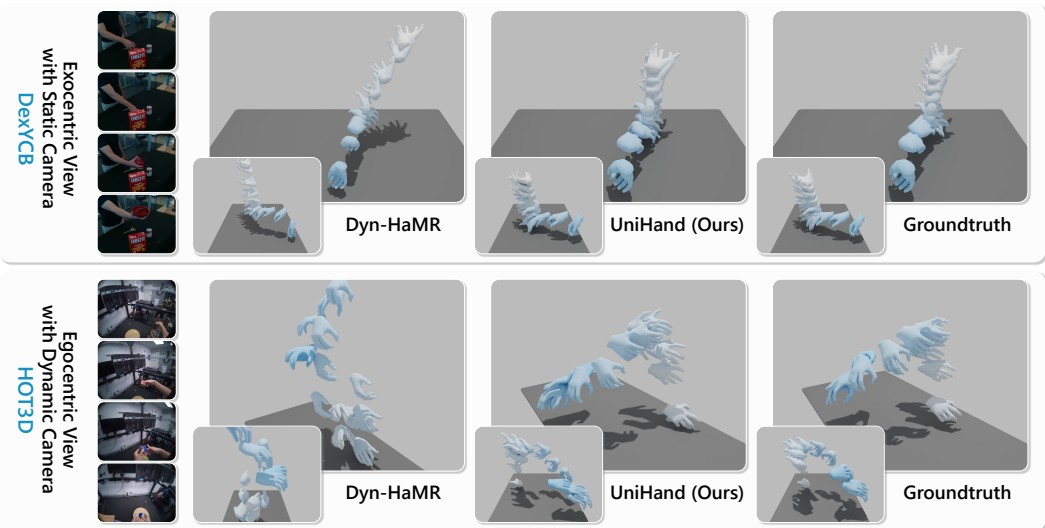

Figure 2: **Visualization of generated hand poses and trajectories.** The first example shows a static camera scenario where *the subject picks up a red bowl*, with significant hand occlusion. The second example is recorded with a dynamic camera, where *the subject picks up and manipulates a magic cube*, involving large hand movements. UniHand produces more accurate hand motion by modeling motions in a canonical coordinate space, even without relying on explicit camera extrinsics.

### 4.3 HAND MOTION IN WORLD COORDINATE SPACE

To evaluate the global consistency of reconstructed hand motions, we conduct experiments in the world coordinate system using the HOT3D dataset, which provides egocentric videos. We consider two categories of methods: camera-space approaches, which estimate hand poses in the camera coordinate system and then transform predictions into the world frame using estimated camera poses from DROID-SLAM (Teed & Deng, 2021), and video-based methods, which jointly infer hand and camera motion in the world space through temporal models.

As shown in Table 3, UniHand consistently outperforms both camera-space and world-space baselines in PA-MPJPE, demonstrating the accuracy of the reconstructed hand poses. Notably, UniHand achieves the lowest G-MPJPE and GA-MPJPE among all camera-space reconstruction methods, despite leveraging explicit camera trajectories estimation for world-space conversion. Our method relies solely on visual observations to model motions in the canonical space. It achieves performance comparable to world-space methods, such as HaWoR and Dyn-HaMR (Yu et al., 2025), that explicitly utilize camera parameters. In addition, UniHand obtains lower acceleration error (AccEr), confirming the temporal smoothness of the reconstructed hand trajectories in the world frame.

We further visualize the generated 3D hand motions in Figure 2. Compared to Dyn-HaMR, UniHand recovers more stable and accurate hand motion sequences, particularly under occlusions or large hand movements. Unlike baseline methods that rely on external SLAM or require per-sequence optimization, UniHand provides a unified and efficient solution for world-space hand motion generation without explicit camera estimation.

### 4.4 ABLATION STUDY

To analyze the effectiveness of the core components and different condition signals, we conduct ablation studies on the DexYCB dataset under the camera coordinate setting and the HOT3D dataset under the world coordinate setting. We also report results on the most challenging occlusion level (75%–100%) of DexYCB. The evaluation metrics follow the same protocol as described in previous experiments.

Table 4: Ablation studies on the core components, design choices, and different condition configurations during inference, evaluated on the DexYCB and HOT3D datasets. Results are reported in terms of MPJPE ($mm$) under different alignment strategies and AUC scores.

| Setups | DexYCB-All | | DexYCB-Occlusion | | HOT3D | | |
|---|---|---|---|---|---|---|---|
| | PA-MPJPE $\downarrow$ | AUC$_J\uparrow$ | PA-MPJPE $\downarrow$ | AUC$_J\uparrow$ | PA-MPJPE $\downarrow$ | G-MPJPE $\downarrow$ | GA-MPJPE $\downarrow$ |
| w/o. Condition Encoders $\mathcal{E}_c$ | 5.21 | 0.895 | 5.56 | 0.889 | 5.92 | 75.49 | 31.03 |
| w/o. Pretrained $\mathcal{E}_{\text{vision}}$ | 6.52 | 0.869 | 6.71 | 0.865 | 8.73 | 146.08 | 39.53 |
| w/o. Hand Perceptron | 7.81 | 0.843 | 8.75 | 0.824 | 12.46 | 180.59 | 48.93 |
| w/o. 3D RoPE | 4.65 | 0.906 | 4.76 | 0.904 | 4.95 | 69.20 | 28.94 |
| w. $c_{\text{vision}}$ | 4.24 | 0.915 | 4.27 | 0.915 | 4.52 | 53.49 | 23.28 |
| w. $c_{\text{2D}}$ | 4.75 | 0.905 | 5.43 | 0.891 | 6.37 | 98.17 | 40.42 |
| w. $c_{\text{3D}}$ | 3.99 | 0.920 | 4.17 | 0.916 | 4.15 | 44.61 | 20.73 |
| w. $c_{\text{vision}}$ and $c_{\text{3D}}$ | 3.48 | 0.931 | 3.67 | 0.926 | 3.82 | 48.11 | 21.36 |
| **Ours (w. $c_{\text{vision}}$ and $c_{\text{2D}}$)** | 4.08 | 0.918 | 4.26 | 0.912 | 4.76 | 63.97 | 25.24 |

**Component Ablation.** The upper part of Table 4 summarizes the ablation results of different components and design choices within the UniHand framework. Setup *w/o.* Condition Encoder $\mathcal{E}_c$ replaces the condition encoders in Joint VAE with an MLP that directly maps condition signals (e.g., 2D keypoints) to the latent dimension. The performance drop indicates that the Joint VAE is critical for learning consistent representations, thereby enabling more effective condition fusion. Setup *w/o.* Pretrained $\mathcal{E}_{\text{vision}}$ uses an identical vision backbone without pretraining. The performance degradation highlights the importance of pretrained visual representations in providing reliable cues for the hand perceptron module. Furthermore, both replacing the hand perceptron module with average pooling over dense vision tokens and replacing 3D RoPE with a standard 1D RoPE lead to clear performance decrease.

**Condition Modality Ablation.** We further evaluate the contribution of each condition modality by testing different inference configurations. As shown in the lower part of Table 4, using only 2D keypoints yields acceptable performance under normal conditions, demonstrating the effectiveness of latent space alignment in the Joint VAE. However, such structural information cannot be reliably extracted under severe occlusions, resulting in poor robustness. Its performance on HOT3D is also limited, indicating that 2D keypoints alone are insufficient for modeling hand motion under dynamic camera movements. Using only $c_{\text{vision}}$ achieves better PA-MPJPE, but its lack of explicit spatial constraints leads to weaker performance in G-MPJPE. The combination of $c_{\text{vision}}$ and $c_{\text{3D}}$ achieves the best overall performance, showing the complementarity between visual evidence and 3D structural cues. However, since 3D keypoints are not directly accessible in real-world scenarios and are mainly applicable to editing tasks, we adopt the $c_{\text{vision}}$ and $c_{\text{2D}}$ configuration for most of our experiments. In practice, 2D keypoints can be easily obtained using pretrained detection backbones, making this setting both effective and practical.

## 5 CONCLUSIONS AND LIMITATIONS

In this work, we introduced UniHand, a unified diffusion-based framework that formulates both hand motion estimation and generation as conditional motion synthesis. UniHand employs a joint variational autoencoder that aligns structured signals such as MANO parameters and 2D skeletons into a shared latent space, ensuring consistency across modalities. In parallel, a hand perceptron module attends to hand-related features extracted from dense tokens of full-size vision inputs, enabling the model to directly exploit rich visual observations without relying on hand detection or cropping. Building on these components, our diffusion-based framework flexibly integrates heterogeneous conditions to generate coherent 4D hand motions. Extensive experiments across multiple benchmarks demonstrate that UniHand achieves robust and accurate hand motion modeling, maintaining strong performance under severe occlusions and temporally incomplete signals. These results highlight the effectiveness of unifying estimation and generation within a single framework, and provide research directions for more general multimodal hand motion modeling in real-world applications.

**Limitations.** UniHand models 4D hand motion directly in the canonical coordinate space without relying on explicit camera extrinsics, thereby providing a unified treatment of both static and dynamic camera scenarios. However, under large camera movements, visual observations or other structured signals alone are insufficient to ensure globally consistent trajectories. This limitation is reflected in our evaluation: while UniHand achieves accurate pose generation and outperforms methods restricted to the camera coordinate space, its global alignment scores remain lower than optimization-based approaches that explicitly leverage camera extrinsics. Future work could incorporate camera estimation into the framework, enabling more accurate trajectory reconstruction under dynamic camera settings.

## ACKNOWLEDGEMENT

This work was supported by the National Natural Science Foundation of China (No. 62472098) and the Science and Technology Commission of Shanghai Municipality (No. 25511106100 and No. 25511104301).

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

## A   METHOD

### A.1   HAND POSE AND CONDITIONS REPRESENTATION

**Canonical Coordinate Space.**   We model 4D hand motion in a canonical coordinate system, defined as the camera space of the first frame. This formulation decouples hand motion from dynamic camera movement, providing a consistent representation across the entire sequence, while remaining applicable to both static and dynamic camera scenarios. In the case of static cameras, the canonical space is identical to the camera space. For dynamic cameras, the camera-to-canonical transformation is computed as:

$$\mathbf{T}^i_{\text{cam}\rightarrow\text{cano}} = [\mathbf{R}^i_{\text{cam}\rightarrow\text{cano}} \mid \mathbf{t}^i_{\text{cam}\rightarrow\text{cano}}]$$
$$= \mathbf{T}^i_{\text{cam}\rightarrow\text{world}} \times \mathbf{T}^1_{\text{world}\rightarrow\text{cam}}, \tag{4}$$

where $\mathbf{T}^i_{\text{cam}\rightarrow\text{world}}$ maps the hand pose from the $i$-th frame camera space to the world space, and $\mathbf{T}^1_{\text{world}\rightarrow\text{cam}}$ maps it back to the camera space of the first frame, which serves as the canonical space.

**Representation.**   A 4D hand motion sequence is denoted as $x = \{x^i\}^N_{i=1}$ of length $N$. Each 3D hand pose $x^i$ is parameterized by the MANO model (Romero et al., 2017b), including hand pose parameters $\Theta^i \in \mathbb{R}^{15\times 3}$, shape parameters $\beta^i \in \mathbb{R}^{10}$, global orientation $\Phi^i \in \mathbb{R}^3$, and root translation $\Gamma^i \in \mathbb{R}^3$. The complete pose $x^i$ is therefore represented in the canonical coordinate space as: $x^i = \{\Theta^i, \beta^i, \Phi^i, \Gamma^i\}$.

The 3D skeleton keypoint condition is obtained by regressing joints from the MANO parameters using the MANO joint regressor $\mathcal{J}$. All joints are transformed into the canonical coordinate space to ensure temporal consistency across the sequence. The 2D skeleton keypoint condition is derived from the projected 3D joints. We preserve the projection defined by the first-frame camera and normalize the coordinates into the range $[0, 1]$ according to the frame resolution, which serves as a consistent visual reference throughout the sequence.

### A.2   JOINT VAE

**Architecture.**   Our Joint VAE adopts a transformer-based architecture. Both the motion encoder $\mathcal{E}$, condition encoders $\mathcal{E}_c$, and the decoder $\mathcal{D}$ are composed of 9 transformer encoder layers. Each layer is configured with a dropout rate of $0.1$, a feed-forward dimension of $2048$, a hidden dimension of $512$, 8 attention heads, and the GELU activation function. The latent space is defined with a dimension of $512$. The autoregressive decoder processes motion in segments of length 8 at a time. We apply Rotary Positional Encoding (RoPE) as temporal positional encoding for the hidden states.

**Losses.**   The Joint VAE is trained with a composed loss defined as:

$$\mathcal{L}_{\text{JointVAE}} = \mathcal{L}_{\text{rec}} + \omega_{\text{KL}}\mathcal{L}_{\text{KL}} + \omega_{\text{latent}}\mathcal{L}_{\text{latent}} + \omega_{\text{aux}}\mathcal{L}_{\text{aux}}. \tag{5}$$

The reconstruction loss $\mathcal{L}_{\text{rec}}$ encourages the reconstructed motion sequence $\hat{x}$ to match the ground-truth motion sequence $x$. It consists of two parts, the MANO parameter reconstruction loss $\mathcal{L}_{\text{mano\_rec}}$ and the joint reconstruction loss $\mathcal{L}_{\text{joint\_rec}}$:

$$\mathcal{L}_{\text{rec}} = \mathcal{L}_{\text{mano\_rec}} + \omega_{\text{joint\_rec}}\mathcal{L}_{\text{joint\_rec}}. \tag{6}$$

The MANO parameter reconstruction loss directly penalizes differences between predicted and ground-truth MANO parameters:

$$\mathcal{L}_{\text{mano\_rec}} = \mathcal{F}_{\text{L1}}(\hat{x}, x), \tag{7}$$

where $\mathcal{F}_{\text{L1}}$ denotes the smoothed L1 loss (Girshick, 2015). The MANO joint reconstruction loss penalizes discrepancies between the 3D joints regressed from the predicted and ground-truth MANO parameters:

$$\mathcal{L}_{\text{joint\_rec}} = \mathcal{F}_{\text{L1}}(\mathcal{J}(\hat{x}), \mathcal{J}(x)), \tag{8}$$

where $\mathcal{J}$ denotes the MANO joint regressor.

The Kullback-Leibler divergence regularization term $\mathcal{L}_{\text{KL}}$  (Kingma & Welling, 2013) regularizes the latent space learned by the Joint VAE by penalizing the divergence between the predicted latent distribution $q(z \mid H)$ and a standard Gaussian $\mathcal{N}(0, I)$ as:

$$\mathcal{L}_{\text{KL}} = KL(q(g \mid x)\|\mathcal{N}(0, \mathbf{I})), \tag{9}$$

where the $KL$ denotes the Kullback-Leibler (KL) divergence. The distribution $q(g \mid x)$ is parameterized by the Gaussian parameters $\mu_g$ and $\sigma_g$. In our implementation, $\mathcal{L}_{\text{KL}}$ is used to avoid arbitrarily high-variance latent spaces of motion global token $g$.

The latent alignment loss $\mathcal{L}_{\text{latent}}$ directly minimizes the distance between the condition latent tokens $z_c$ (from the Condition Encoders) and the motion latent tokens $z$ (from the Motion Encoder). This encourages the information encoded from the two different modalities to align in the shared latent space. Including 2D condition encoder and 3D condition encoder alignment constraints:

$$\mathcal{L}_{\text{latent}} = \mathcal{L}_{\text{latent\_2D}} + \mathcal{L}_{\text{latent\_3D}}, \tag{10}$$

$$\mathcal{L}_{\text{latent\_c}} = \mathcal{F}_{\text{MSE}}(z_c, z), \tag{11}$$

where $\mathcal{F}_{\text{MSE}}$ denotes the mean squared error (MSE) loss.

The auxiliary loss $\mathcal{L}_{\text{aux}}$ regularizes predicted motion $\hat{x}_c$ reconstructed from condition latent $z_c$

$$\mathcal{L}_{\text{aux}} = \mathcal{L}_{\text{aux\_2D}} + \mathcal{L}_{\text{aux\_3D}}. \tag{12}$$

$$\mathcal{L}_{\text{latent\_c}} = \mathcal{F}_{\text{L1}}(\hat{x}_c, x). \tag{13}$$

### A.3 LATENT DIFFUSION MODEL

**Architecture.** The condition denoiser $\mathcal{G}_\theta$ is implemented as a transformer-based architecture consisting of 16 transformer layers as illustrated in Figure 1. Each layer is configured with a feed-forward dimension of 2048, a hidden dimension of 512, 16 attention heads, and the GELU activation function. The latent space has a dimensionality of 512, consistent with the Joint VAE. Following Yang et al. (2024), the diffusion timestep $t$ is injected into the network through the modulation module of an adaptive LayerNorm. For temporal modeling, we apply Rotary Positional Encoding (RoPE) as temporal positional encoding to the hidden states. For vision encoding, we adopt the pretrained DINO-v2 Oquab et al. (2023) backbone, with weights kept frozen.

**3D RoPE.** We adopt Rotary Positional Encoding (RoPE) (Su et al., 2024), which has been shown to improve scalability and adaptability. RoPE encodes relative positional information through rotations in the complex space:

$$R_i(x, m) = \begin{bmatrix} \cos(m\theta_i) & -\sin(m\theta_i) \\ \sin(m\theta_i) & \cos(m\theta_i) \end{bmatrix} \begin{bmatrix} x_{2i} \\ x_{2i+1} \end{bmatrix}, \tag{14}$$

where $x$ is the input query or key representation, $m$ is the positional index, $i$ is the feature dimension index, and $\theta_i$ is the frequency.

Given that the vision backbone extracts tokens $v$ with temporal length $N$, spatial height $h$, and width $w$, and to capture both spatial and temporal structures, we extend RoPE into a 3D formulation, following prior work (Kong et al., 2024; Yang et al., 2024). The attention dimension is divided into three complementary subspaces, each dedicated to one axis. Independent sinusoidal embeddings are generated for the temporal, horizontal, and vertical dimensions, capturing relative positional information along each axis. Concretely, we compute the rotary frequency matrix separately for the coordinates of time, height, and width. The feature channels of the query and key are partitioned into three segments $(d_t, d_h, d_w)$, and each segment is multiplied by the corresponding coordinate frequency. The outputs are then concatenated to produce position-aware query and key embeddings, which are applied in attention computation. Compared to the standard RoPE, this 3D extension jointly encodes temporal continuity and spatial structure in a unified representation.

**Losses.** The denoiser model is trained with the following losses:

$$\mathcal{L}_{\text{denoiser}} = \mathcal{L}_{\text{simple}} + \omega_{\text{rec}}\mathcal{L}_{\text{rec}}. \tag{15}$$

We train the denoiser to predict the clean latent variable with the simple objective $\mathcal{L}_{\text{simple}}$. Training proceeds by sampling $z_0$ from the dataset, applying the forward process to obtain a noisy latent $z_t$, predicting $\hat{z}_0$ using $\mathcal{G}_\theta$, and minimizing the reconstruction error. The simple objective is defined as:

$$\mathcal{L}_{\text{simple}} = \mathbb{E}_{(z_0, C) \sim q(z_0, C), t \sim [1, T], \epsilon \sim \mathcal{N}(0, \mathbf{I})} \mathcal{F}_{\text{MSE}}(\mathcal{G}_\theta(z_t, t, C), z_0), \tag{16}$$

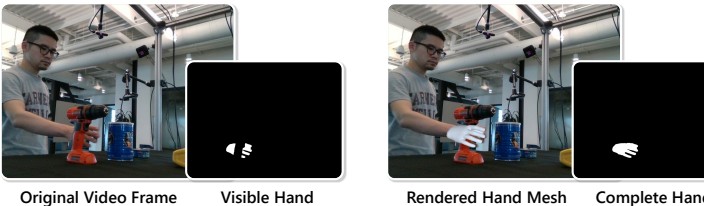

Original Video Frame     Visible Hand          Rendered Hand Mesh     Complete Hand

Figure 3: Illustration of hand occlusion level computation on the DexYCB dataset.

where $\hat{z}_0 = \mathcal{G}_\theta(z_t, t, C)$ denotes the predicted clean latent, and $\mathcal{F}_{\mathrm{MSE}}$ is a distance function which is implemented using the mean squared error (MSE) loss.

The reconstruction loss $\mathcal{L}_{\mathrm{rec}}$ (same as defined in Eq. (6)) encourages the predicted motion sequence $\hat{x}$ to remain close to the ground-truth sequence $x$ by jointly penalizing discrepancies in both MANO parameters and the regressed 3D joints.

**Inference.** At inference time, we initialize with Gaussian noise $z_T \sim \mathcal{N}(0, I)$. The denoiser is applied iteratively, where at each step it predicts the clean latent $\hat{z}_0$ and updates the noisy latent $z_t$ towards a lower-noise state, until a clean latent $z_0$ is obtained. The final latent $z_0$ is then decoded by the autoregressive decoder in the Joint VAE to generate a hand motion sequence $\hat{x}$.

Benefiting from the design of the Joint VAE, structured control signals such as 2D and 3D keypoints are encoded into the shared latent space and can be directly fused with the noisy latent $z_t$. Visual information is extracted by the frozen vision backbone, processed through the hand-relevant attention module, and represented as hand tokens, which are integrated into the denoiser at each step. We further adopt classifier-free guidance (CFG), assigning an independent unconditional token to each control modality. This design enables flexible integration and combination of different condition inputs.

## B    EXPERIMENTAL SETUP

### B.1    DATASETS

We train our model on the DexYCB (Chao et al., 2021) and HOT3D (Banerjee et al., 2025) datasets, and additionally evaluate out-of-domain generalization on HO3D (Hampali et al., 2020). To simplify learning, we horizontally flip input images and corresponding annotations whenever the targeted hand is left, resulting in a right-hand-only network. Unless otherwise specified, UniHand is trained exclusively on the training splits of DexYCB and HOT3D, and all reported results are obtained from a single unified checkpoint, without dataset-specific fine-tuning or architectural modifications.

Since both DexYCB and HOT3D contain motion sequences, during training, we randomly select a valid initial pose within a sequence and sample consecutive frames to construct motions of length $N = 48$. At the inference stage, the sequence length is required to be an integer multiple of the autoregressive decoding segment length. If this condition is not satisfied, we pad the sequence by repeating the control conditions of the final frame.

**DexYCB.** DexYCB (Chao et al., 2021) is a large-scale dataset containing $8,000$ videos of single-hand object manipulation. It features 10 subjects performing grasps on 20 objects from the YCB-Video dataset (Xiang et al., 2018). Each action sequence is captured by 8 synchronized RGB-D cameras from a fixed third-person viewpoint. For evaluation, we follow the official protocol and adopt the default split (S0) for training and testing.

To evaluate the degree of hand occlusion, we compute the ratio between the occluded hand region and the complete hand region. As illustrated in Figure 3, we obtain two types of masks: the visible hand mask $M_{\mathrm{vis}}$, where only the non-occluded pixels of the hand are labeled as 1 (provided by the dataset), and the complete hand mask $M_{\mathrm{hand}}$, which is obtained by decoding MANO parameters and rendering the hand mesh, covering the entire hand region regardless of occlusion. Formally, the

occlusion ratio is defined as:

$$r_{\text{occ}} = \frac{|M_{\text{hand}}| - |M_{\text{hand}} \cap M_{\text{vis}}|}{|M_{\text{hand}}|}, \qquad (17)$$

where $|M|$ denotes the number of pixels labeled as $1$ in mask $M$. This metric allows us to categorize frames in the DexYCB dataset into different occlusion levels.

**HOT3D.** HOT3D (Banerjee et al., 2025) is a first-person dataset recorded with dynamic cameras, covering both single-hand and two-hand manipulations. It provides ground-truth camera trajectories as well as world-coordinate MANO annotations for each frame.

In our experiments, we use the HOT3D-Clips version, which consists of carefully selected subsequences from the original dataset. Each clip contains roughly 150 frames, corresponding to about 5 seconds of video. We adopt the subset collected with the Aria device and use only the main-view RGB images as vision conditions, since the Quest3 device does not provide RGB data. Ground-truth poses are available for every modeled object and hand in all frames. Because the official test split does not provide ground-truth annotations, we use the split based on the official training set, resulting in $1,272$ clips for training and $244$ clips for testing.

### B.2 Implementation Details

All experiments are conducted on 4 NVIDIA 80GB H800 GPUs. We adopt DeepSpeed (Rasley et al., 2020) for training to reduce memory consumption and improve efficiency. The AdamW (Loshchilov & Hutter, 2017) optimizer is used with an initial learning rate of $1 \times 10^{-4}$, scheduled with 100 warmup iterations followed by linear annealing.

We first train the Joint VAE. A small KL weight $\omega_{\text{KL}} = 1 \times 10^{-4}$ is applied to maintain an expressive latent space while preventing arbitrarily high-variance latent variables. The other loss terms are balanced with weights of $\omega_{\text{joint\_rec}} = 0.5$ for the joint reconstruction loss, $\omega_{\text{latent}} = 0.1$ for the latent loss, and $\omega_{\text{aux}} = 0.1$ for the auxiliary loss. After training, the motion encoder, condition encoders, and autoregressive decoder are frozen. The latent denoiser is then trained using DDPM (Ho et al., 2020) with 50 diffusion steps and a cosine noise scheduler. A weight of $\omega_{\text{rec}} = 1.0$ is applied during training.

At inference time, we employ DDIM (Song et al., 2020) with 10 diffusion steps for efficient generation while mitigating error accumulation, and set the CFG scale to $\omega = 2$. Following the ablation study, we adopt vision frames and 2D keypoints as the default condition configuration, since 3D keypoints are not directly available in real-world scenarios. For 2D keypoint detection, we utilize the pre-trained ViT backbone from HaMeR (Pavlakos et al., 2024), which is also employed for the initialization of the first-frame hand pose.

## C Visualization

### C.1 Hand Motion in Camera Coordinate Space

To further demonstrate the effectiveness of our method under challenging scenarios such as severe occlusions and temporally incomplete conditions, we present qualitative comparisons in Figure 5, Figure 6, and Figure 7. We compare HaMeR (Pavlakos et al., 2024) with our proposed UniHand. The visualizations show that UniHand reconstructs more temporally stable and geometrically plausible hand poses, particularly when the hand is heavily occluded or interacting with objects. These results indicate that our unified generative framework effectively leverages heterogeneous conditions to maintain robustness and fidelity in complex real-world scenarios.

### C.2 Hand Motion in World Coordinate Space

In Figure 4, we include additional visualizations of generated hand poses and trajectories in world coordinates. The first example presents a left-hand motion sequence, which illustrates how UniHand maintains consistent predictions across both hands. As described in the main text, UniHand horizontally flips input images and their corresponding annotations whenever the targeted hand is

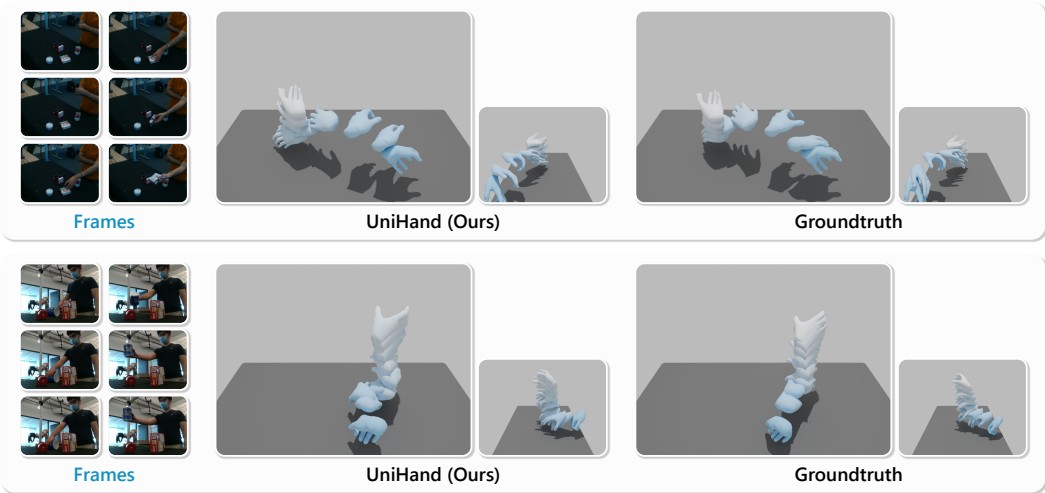

Figure 4: Additional visualization of generated hand poses and trajectories.

left, resulting in a right-hand-only network that simplifies learning. Thus, the model always predicts right-hand MANO parameters, which is also a standard practice adopted by prior methods such as HaMeR. For left-hand inputs, we invert the flipping transformation on the predicted right-hand MANO parameters to obtain the corresponding left-hand result.

## D STATEMENT

### D.1 THE USE OF LARGE LANGUAGE MODELS (LLMS)

We used Large Language Models (LLMs) only as a writing assistant for language polishing during the preparation of this paper. LLMs were not used in the ideation, experiments, data collection, or result analysis. The authors take full responsibility for the content of this paper, including the text that was refined with the assistance of LLMs.

### D.2 REPRODUCIBILITY STATEMENT

We have taken several steps to ensure the reproducibility of our work. In the supplementary material, we provide the core code for the proposed method, data loader, and inference pipeline. A detailed description of dataset preprocessing, splits, and statistics is included in Appendix B.1. Comprehensive model architectures and implementation details are presented in Appendix A and B.2. These materials, together with the released code, are intended to facilitate the reproduction of our results and further research on this topic.

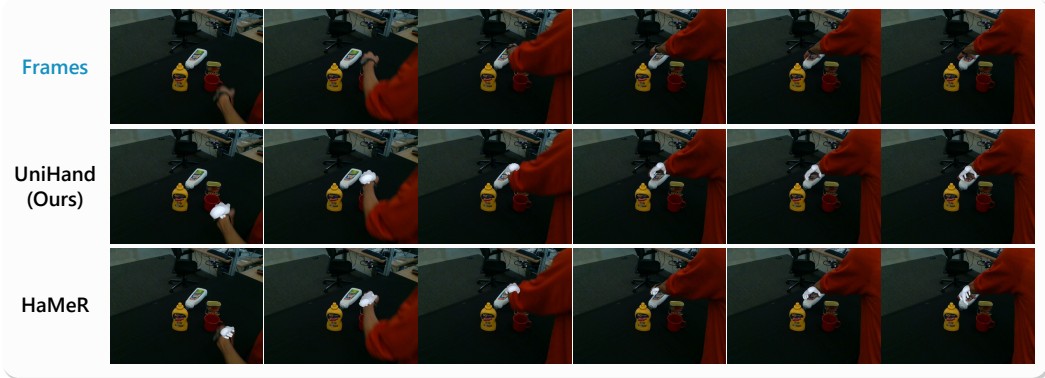

Figure 5: Qualitative comparison between HaMeR and our UniHand. Our method generates more continuous and accurate hand pose sequences compared to HaMeR.

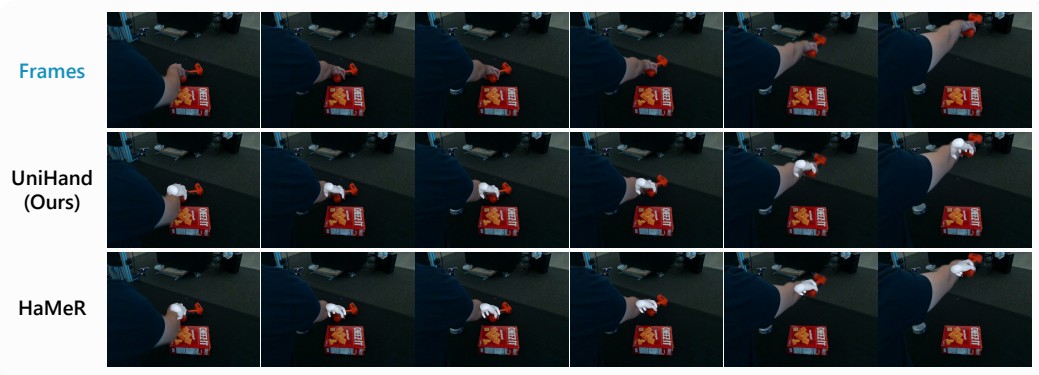

Figure 6: Qualitative comparison between HaMeR and our UniHand. In cases of severe hand self-occlusion, HaMeR misclassifies the right hand as the left hand, resulting in poor reconstruction quality, whereas UniHand generates reliable and consistent hand motions.

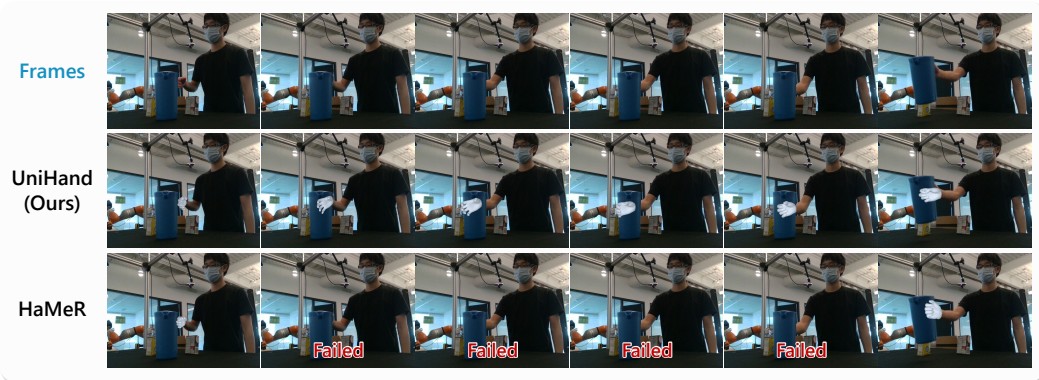

Figure 7: Qualitative comparison between HaMeR and our UniHand. HaMeR fails to estimate valid poses in video frames where the hand is absent, whereas UniHand maintains stable reconstructions by exploiting vision perception and temporal modeling.

