# OpenReview forum: "UniHand: A Unified Model for Diverse Controlled 4D Hand Motion Modeling"
_ICLR.cc/2026/Conference — ICLR 2026 Poster_

### Official Review · Reviewer_4cS2 · 2025-10-15

**Soundness:** 3
**Presentation:** 3
**Contribution:** 3
**Rating:** 6
**Confidence:** 3

**Summary:**

This paper presents UniHand, a unified diffusion-based framework for 4D hand motion modeling that jointly addresses motion reconstruction and generation. It reformulates both tasks as a conditional motion synthesis problem, bridging the gap between estimation and generation. A Joint Variational Autoencoder (Joint VAE) aligns heterogeneous conditional inputs such as 2D and 3D keypoints and MANO parameters into a shared latent space, enabling robust modeling under incomplete or inconsistent data. The Hand Perceptron module extracts hand-specific features directly from full visual inputs without requiring explicit detection or cropping, while a canonical coordinate system ensures spatial and temporal consistency under dynamic camera settings. Experiments on DexYCB, HO3D, and HOT3D datasets demonstrate that UniHand achieves state-of-the-art results, especially under severe occlusion or partial input, validating its effectiveness and generalization for unified hand motion estimation and generation. However, despite its strong performance, the method suffers from algorithmic complexity and high computational cost, and the exploration of the unique challenges in multi-modal alignment for 4D hand pose estimation remains insufficient.

**Strengths:**

- The paper proposes a unified diffusion-based framework that integrates both estimation and generation for 4D hand motion modeling, offering a fresh formulation of conditional motion synthesis that extends beyond task-specific designs.
- The technical design, including the Joint VAE and Hand Perceptron modules, is well-motivated and validated through comprehensive experiments on multiple datasets, showing robustness under occlusion and dynamic camera motion.
- The paper is clearly written and systematically structured.

**Weaknesses:**

- Real-world deployment may be limited without efficient preprocessing of input modalities.
- Heavy computational and data requirements for training.

**Questions:**

- While UniHand demonstrates strong performance, the diffusion-based generation pipeline and multimodal latent alignment introduce high computational cost. Can the authors quantify the training and inference time compared to existing methods, and discuss possible simplifications or acceleration strategies?
- The paper acknowledges that UniHand struggles to maintain globally consistent trajectories under large camera movements due to the lack of explicit camera extrinsics. Could the authors elaborate on potential ways to address this—e.g., by incorporating implicit camera modeling or scene-aware constraints?
- Although UniHand integrates visual, 2D, and 3D conditions through a joint VAE, the paper could further investigate how multimodal alignment contributes to 4D hand pose estimation quality. Would ablation or visualization of the latent space help clarify how modalities interact and which provide the most benefit?

---

> ### Author Response · Authors · 2025-11-20
> **Response to Reviewer 4cS2**
>
> We sincerely thank the reviewer for the constructive feedback and for recognizing the formulation novelty, technical soundness, and experimental strengths of UniHand. Below, we address the specific questions and concerns in detail.
>
> ---
>
> **1. Computational cost and acceleration strategies.**
>
> Diffusion-based generation methods typically incur additional computation cost due to their multi-step denoising process. However, UniHand is designed with several optimizations that substantially reduce this cost. First, the condition encoder is executed only once, and its output remains fixed throughout the denoising process. Second, we adopt DDIM sampling with only 10 denoising steps, which significantly accelerates inference. Third, although the latent decoder is autoregressive, it processes motion segments of length 8.
>
> To better illustrate UniHand’s computation cost, we compare its inference cost with two representative baselines: HaMeR, an image-level hand pose estimation method, and Dyn-HaMR, a multi-stage hand motion estimation pipeline. Despite the iterative denoising steps, UniHand generates an entire sequence in a single multi-step diffusion process, whereas image-level models such as HaMeR must perform inference for every frame. Dyn-HaMR pipelines include substantial multi-stage cost, including per-frame pose initialization, temporal infilling, SLAM-based camera trajectory estimation, and global trajectory optimization. UniHand avoids these components (including crop preprocessing and camera parameter estimation) and replaces them with a single conditional generative model. A comparison of the inference cost for 48 frames is shown below.
>
> - HaMeR: 23.8s, including (1) 21.7s: detect hand and crop image, (2) 2.1s: per-frame inference
> - Dyn-HaMR: 70.5s, including (1) 22.9s: per-frame initialization, (2) 23.0s: infilling and camera estimation, (3) 24.6s: global optimization
> - UniHand (Ours): 21.2s, (1) optional 19.8s: detect 2D keypoints, (2) 1.4s: diffusion and autoregressive decoding
>
> On the training side, the cost is also moderate. HaMeR trains with an effective batch size of 1024 for 420k iterations. In contrast, UniHand trains the Joint VAE for 50k iterations with a batch size of 32, followed by the diffusion model for 200k iterations, resulting in a total cost comparable to or lower than that of existing methods.
>
> Finally, we note several promising directions for further acceleration.
> The diffusion backbone can benefit from the distillation method. In addition, emerging adaptive-step strategies can further reduce sampling cost based on the current denoising state. These techniques can be integrated into UniHand, and we consider them promising directions for future work.
>
> **2. Handling global consistency under large camera motion.**
>
> Thank you very much for the insightful suggestion. Incorporating implicit camera modeling is indeed a promising direction to further improve global trajectory consistency. One potential approach is to introduce an optional camera extrinsics input. When camera extrinsics are available, the model can directly leverage them to obtain more stable world-level trajectories. When camera parameters are unavailable, the model can instead rely on a learnable latent token. During training, this token can be supervised using available camera extrinsics, enabling the model to approximate camera motion implicitly at inference time. We consider this a promising extension of our canonical-space formulation and plan to explore its effectiveness in future work.
>
> **3. Contribution of multimodal alignment**
>
> Thank you for raising this important point. We agree that examining how multimodal alignment affects 4D hand motion estimation is valuable. In addition to the ablation already included in the main paper, we further define two additional alignment configurations to clarify the contribution of individual modalities in the Joint VAE.
>
> - Setup1: Joint VAE trained with hand pose only (corresponding to *w/o. condition encoders* in Figure 4).
> - Setup2: Joint VAE trained with hand pose and 2D keypoints.
> - Setup3: Joint VAE trained with hand pose and 3D keypoints.
>
> We evaluate PA-MPJPE and AUC on DexYCB:
>
> |  | PA-MPJPE | AUC |
> | --- | --- | --- |
> | Setup1 (pose only) | 5.21 | 0.895 |
> | Setup2 (pose and 2D) | 5.09 | 0.897 |
> | Setup3 (pose and 3D) | 4.40 | 0.912 |
> | Ours (all modalities) | 4.08 | 0.918 |
>
> These results show that multimodal alignment plays a central role in improving motion estimation performance. 3D keypoints provide the strongest alignment supervision. 2D keypoints still provide useful alignment, and are particularly valuable in practical scenarios where reliable 2D detections are easier to obtain. We include this ablation in **Appendix D** for clarity.
>
> ---
>
> We thank the reviewer again for the positive assessment and the valuable suggestions. We have revised the paper to include the additional clarifications and resubmitted the PDF file.

---

> > ### Comment · Reviewer_4cS2 · 2025-11-26
> >
> > Thanks for your response! Authors generally addressed my concerns. Although there is a little computational burden, it is acceptable, and I will keep my rating! Please carefully address the issues raised by Reviewer `LJ9z`.

---

### Official Review · Reviewer_LJ9z · 2025-10-28

**Soundness:** 1
**Presentation:** 3
**Contribution:** 2
**Rating:** 2
**Confidence:** 4

**Summary:**

This paper aims to unify both hand pose estimation and generation using a single diffusion-based framework.
The proposed model, UniHand, handles both structured MANO and key points (as conditions for generation), as well as images (for estimation).
Separate encoders are used to bring different hand conditions into the same latent space, then the method runs denoising diffusion process in the latent space, aiming to to produce smooth 4D motion output.

**Strengths:**

This paper is well motivated: it aims to bring the estimation and the generation model together.
The proposed model indeed can do these two task in a unified way.
This paper does not propose a new problem formulation, but the effort to introduce an unified solution for two problems are interesting and valuable.

The writing is clear in general: we can understand how the proposed framework achieve the proposed goal at the high level.

**Weaknesses:**

The main issue of this paper is it does not address the proposed goal: unifying both estimation *and generation*. The generation ability of the framework is not tested/reported.
The paper is motivated by unifying the estimation and generation into the same framework. The motivation is sound, and the de-noiser in the framework is indeed a generative model. However, all results are reported **only on the hand pose estimation task**.
The sole focus of estimation deviates from what is described in the title, abstract and intro, as well as in the related works where the authors talk about unifying generation under their framework in Line 134-136.
The authors could have shown their results on any generation problems, e.g. [Zuo et al., 2023] in Line 128 or [Zuo et al., 2024] in Line 115-116 of the grasp generation problem. I understand that extending the system for these generation tasks can involves much more work, but since the authors claims the unified method, one would expect the generation task results.

In Line 347-348, the authors proposes to evaluate the generation via estimation:
> Hand pose estimation in the camera coordinate space provides the most direct way to evaluate the quality of motion generation conditioned on visual observations.

but above is saying that the estimation and the generation are the same task, which contracts what the authors say in Line 013 and throughout the whole introduction!

The author should really test their framework on a few tasks listed in their Sec 2.2, without the results on the generation task, otherwise this paper gives a wrong impression to the audience.

Regarding the estimation evaluation setup, it is unclear to me when $c_{2D}$ and $c_{3D}$ are provided to the model? In my understanding, the standard estimation task setup does not provide $c_{2D}$ and $c_{3D}$ to the model.

Since providing experiment results on the real generation task will lead to substantial changes to the paper and is beyond what can be done during rebuttal, I suggest rejection.

**Questions:**

Apart from the mismatch between what is claimed and what is experimented, the writing of this paper is good in general.

The points that is unclear:

1. Figure 1, bottom left, what are two red boxes represent at the bottom left? are they $g$?

2. Shouldn't motion encoder  be called hand pose encoder?

3. Line 207-208, better to say "At each _autoregression_ step" instead of simply "At each step", which can mean diffusion step.

4. Line 883, DeepSpeeds need reference.

5. Line 118, EASY-HOI is not relevant to Hand Motion Generation.

---

> ### Comment · Reviewer_LJ9z · 2025-11-13
> **More justification for the 2 rating**
>
> I read my review again.
> I would like to put some comments on my rating of score 2.
> The "Soundness: 1. poor" and rating of "2 clear reject" do not mean the method is not good; instead, I mean there is a flaw in how the authors present the overall setting, i.e. the authors claim the method unifies the generation task while not showing any results on generation, e.g. grasp synthesis.
>
> Within the hand pose estimation setting, the method is actually well motivated and experiments look sensible.
>
> The reason I rate as 2 is simply because there is such a big hole between what is claimed and what is experimented.
>
> The fix is simple in principal: the authors can either change their claim from "unifying generation" to "adopting generation appraoch", or by showing generation results. However, I understand that doing either way will lead to substantial changes to the draft, which does not seem to fit the purpose of rebuttal, that is why I suggest a reject right at the beginning.
>
> I'm happy to hear authors' defence or any point I have misunderstood.

---

> ### Author Response · Authors · 2025-11-20
> **Response to Reviewer LJ9z (part1)**
>
> Thank you for your thoughtful review and for acknowledging the motivation of our framework. We fully agree that clear alignment between claims and experiments is crucial. However, we **respectfully disagree** with your statement that *“there is such a big hole between what is claimed and what is experimented”* and that “*the generation ability of the framework is not tested/reported”*. Since this is the main concern underlying your overall rating, we first focus on clarifying this issue.
>
> In what follows, we provide a concise clarification of (A) what our “unification” claim means, (B) what types of generation are in the scope of our unification, and (C) how our experiments evaluate the generative capabilities.
>
> ---
>
> **A. What does our “unification” claim mean?**
>
> We refer specifically to a formulation-level unification: **both estimation and generation are formulated as conditional motion synthesis**. We do not claim to unify or cover all existing hand-related generation tasks.
>
> Concretely, Sec 3.1 explicitly describes that UniHand “formulates hand motion estimation and generation within a unified framework of conditional hand motion generation,” where both tasks aim to synthesize a hand motion sequence $x$ conditioned on heterogeneous signals $C$ (vision, 2D/3D keypoints, MANO parameters, and temporal masks).
>
> Thus, our unification refers to using a **single conditional generative diffusion model**, operating in **one shared latent motion space**, capable of **supporting different condition modalities and different levels of temporal completeness** within the same formulation. This is a modeling unification, not a claim of universal task coverage.
>
> ---
>
> **B. What types of generation are in the scope of our unification?**
>
> Within our formulation, generation refers to synthesizing hand motion from heterogeneous and potentially incomplete conditions. This includes the following instances in our paper:
>
> (1) Image-conditioned synthesis (traditionally viewed as estimation): generating a motion sequence given RGB video inputs;
>
> (2) Keypoint-conditioned synthesis: generating motion from 2D/3D keypoints or their combinations;
>
> (3) Motion completion under occlusion: generating occluded or missing segments using partial temporal conditions;
>
> (4) Dynamic-camera motion synthesis: generating globally coherent motion in dynamic camera scenarios without camera extrinsics.
>
> In contrast, we **do not** claim that UniHand performs object-centric HOI grasp generation. As discussed in related work (Sec 2.2), we treat HOI methods as a different line of work, since they rely on object meshes/poses, or task-specific interaction pipelines. Our condition set $C$ does not include object geometry or 6D object poses, and we do not target this class of tasks. We will further clarify this to avoid potential misinterpretation.
>
> ---
>
> **C. How do our experiments evaluate the generative capabilities?**
>
> UniHand always synthesizes motion by sampling from a diffusion process in the latent motion space, even estimation benchmarks are solved through **conditional motion generation**, not regression. Specifically, we evaluate the generative capability of UniHand through multiple forms of conditional generation:
>
> (1) Image-conditioned generation (Table 2)
>
> This task is traditionally benchmarked as pose estimation. We use the estimation benchmarks only because they are the established protocol for evaluating **image-conditioned motion synthesis**, not because UniHand is limited to estimation.
>
> (2) Motion completion under occlusion (Table 1)
>
> Occlusions frequently cause temporal incompleteness. Traditional estimation pipelines (e.g., HaWoR) rely on an additional infilling module to complete missing frames. In our framework, this becomes a **generation task with temporally incomplete conditions.**
>
> (3) Dynamic-camera motion synthesis (Table 3)
>
> For sequences with dynamic camera motion, existing pipelines require a separate camera-extrinsic estimator and a global optimization module to produce globally consistent trajectories. UniHand instead performs canonical-space motion generation using a single conditional generative model.
>
> (4) Keypoint-conditioned synthesis (Table 4)
>
> We vary the conditioning signals, using only 2D keypoints, only 3D keypoints, or their combinations. These results demonstrate the model’s ability to perform multimodal conditional generation and validate the consistency of the unified latent motion space.
>
> These evaluations cover the full range of conditional generation tasks defined within our unified formulation.
>
> ---
>
> We also note that the other reviewers recognized our approach as a unified framework for motion estimation and generation. We hope that the step-by-step clarification above addresses your main concern. If there is any specific part of this reasoning that remains unclear or unconvincing, we would be very grateful to hear it and would be happy to further clarify.

---

> ### Author Response · Authors · 2025-11-20
> **Response to Reviewer LJ9z (part2)**
>
> In addition to the main concern discussed above, we sincerely appreciate the additional questions and suggestions. We address them one by one below.
>
> ---
>
> **1. Interpretation of the two red boxes in the bottom left of Figure 1.**
>
> Thank you for pointing this out. The two red boxes with white fill correspond to the learnable distribution tokens $T_{\mu}$ and $T_{\sigma}$, which are fed into the encoder transformer. As described in Line 203, the encoder predicts Gaussian parameters $\mu_g, \sigma_g$, from which the global motion token $g$ is sampled. We will revise the figure to explicitly label these tokens and avoid any ambiguity.
>
> **2. Naming of the motion encoder.**
>
> Our use of “motion encoder” is intentional to distinguish its role from that of the condition encoder. Within the overall conditional generation framework, the motion encoder processes noisy representations of the target motion to be generated, while the condition encoder processes the condition signals. The distinction lies in their functional roles in the generative pipeline, rather than in any specific input modality.
>
> **3. “At each step” in Lines 207–208.**
>
> Thank you for the helpful suggestion. We will revise the wording to “at each autoregression step” to prevent confusion with diffusion time-steps.
>
> **4. Missing reference for DeepSpeed in Line 883.**
>
> Thank you for pointing out this issue. We will add the citation to DeepSpeed.
>
> *Rasley, Jeff, et al. Deepspeed: System optimizations enable training deep learning models with over 100 billion parameters.*
>
> **5. Relevance of EASY-HOI**
>
> We appreciate this comment. We agree that EASY-HOI is not directly relevant to hand motion generation and may cause confusion in the current context. We will remove the reference to avoid any implication of a connection to our generation tasks.
>
> ---
>
> We hope these clarifications address all remaining questions. We welcome any further discussion, and we will revise the paper accordingly and resubmit the PDF file before November 24.

---

> > ### Comment · Reviewer_LJ9z · 2025-11-24
> >
> > Suggestion::
> >
> > 1. There are two names in Figure-1: Stage-1 Autoregressive Decode and Stage-2 AutoReg Decoder, pick one to keep consistency.
> >
> > 2. State “two stage” training somewhere in the main text and refer readers to B.2. At this point the readers need to look for the training implementation to understand the training is done in separate stages. Even though this is standard.

---

> ### Comment · Reviewer_LJ9z · 2025-11-24
>
> # The technical design
>
> First, I thank the authors for clarifying the formulation-level unification, and that no task coverage is claimed on the grasp generation for Hand-Object Interaction. I also thank the authors for restating that the pose estimation is performed within the generative framework.
>
> Given their now narrowed claim, I read the draft again. I acknowledge the effectiveness of Joint VAE and conditional denoiser, I update the score to 4 to reflect this.
>
> Unfortunately, I still haven’t fully understand the Classifier-free guidance starting from Line301, I’m not sure how it fits into the framework.
>
> # Motivation still needs justification
>
> Currently in the main paper, the authors have:
>
> Line 016:
> > “Generation approaches focus on infilling motion from incomplete sequences and synthesizing diverse motions from inputs such as object priors”
>
> Line 043-036
> > “Generation approaches, on the other hand, focus on infilling motions from incomplete sequences and synthesizing diverse motions from structured inputs such as object meshes or 6D object pose “
>
> where the authors initially use _infilling_ and _motion from object_ as two examples for unification.
> Since the authors now clarified that the unification is at formulation-level and do not claim to cover those tasks, I believe these two examples are no longer appropriate, because i) there is no reference work given on“infilling motion” and ii) object-related task is irrelevant. As a result, the benefit of unifying “generation” is not grounded by any examples. Bringing up those two examples in fact causes misunderstanding. I’m wondering how the authors will deal with this? What is the authors' resolution to the two invalid examples?
>
> ##
>
> > [From Author Reply] B. What types of generation are in the scope of our unification?
>
> The authors can of course redefine the term of generation using their proposal. But these tasks should be clairified, to not be confused with the standard generation task that do not have a corresponding ground-truth, e.g. generating a sequence of hand sign language.

---

> ### Author Response · Authors · 2025-11-25
> **Response to Reviewer LJ9z**
>
> We are glad that the clarification on formulation-level unification addressed your main concerns. We respond to the remaining questions point by point below.
>
> **1. Clarification on CFG.**
>
> Thank you for raising this question. As described in the main text, to further enhance condition flexibility, we adopt classifier-free guidance (CFG). Unlike conventional CFG settings, hand motion and other control modalities (e.g., 2D keypoints) do not possess natural unconditional forms $c_\varnothing$. To address this, we introduce an independent learnable unconditional token for each control modality.
> During training, with a predefined probability $p$, we replace the valid condition with its corresponding learnable unconditional token.
> This teaches the model the unconditional branch for each modality.
> During inference, we directly use these learned unconditional tokens as the modality-wise $c_\varnothing$ for CFG.
>
> **2. Justification of the Motivation.**
>
> In the revised version, we have updated both the abstract and the Introduction to avoid any implication that we target HOI grasp generation or object-based synthesis. The related sentences have been rewritten as:
>
> *“Generation approaches, on the other hand, focus on synthesizing hand poses by exploiting generative priors under multi-modal structured inputs, such as 2D and 3D skeletons, and infilling motions from incomplete sequences.”*
>
> This aligns with our definition of conditional hand motion synthesis, avoids references to object priors, and we have added citations for the relevant generation tasks accordingly. The specific revisions are available in the revised PDF file.
>
> **3. Revision of Figure 1.**
>
> Thank you for the helpful suggestion. We have unified the terms “Autoregressive Decoder” and “AutoReg Decoder” into a single, consistent name throughout Figure 1, as reflected in the updated PDF.
>
> **4. Two-Stage Training Strategy.**
>
> We agree that stating the two-stage training strategy in the main text improves clarity. We have now made this explicit around Line 300 and added a direct reference to Appendix B.2, where the full details are provided.

---

> > ### Comment · Reviewer_LJ9z · 2025-11-25
> >
> > I thank the authors for updating the draft.
> >
> > The clarification on the generation approach addressed my concerns of the motivation.
> >
> > Regarding the scope of this work, the data used for learning the Joint-VAE latent representation currently come solely from hand–object interaction sequences where objects are present. Are non-object (i.e., pure hand motion) sequences considered out of scope? The framing of “4D hand motion” seems to imply that this work should also encompass general hand motions without objects.

---

### Official Review · Reviewer_76ft · 2025-11-01

**Soundness:** 3
**Presentation:** 4
**Contribution:** 3
**Rating:** 8
**Confidence:** 4

**Summary:**

The work seeks to unify the hitherto distinct domains of hand motion generation and hand motion reconstruction. This is achieved by a variational autoencoder embedding various non-visual modalities (hand trajectories, human body keypoints) into a shared latent space. In parallel, a hand perceptron extracts frame-wise features from visual inputs. The extracted non-visual latent representations are then used by a latent diffusion model to generate the hand pose sequence, with visually informative features being provided at every layer of the motion denoiser. The work compares the proposed method with multiple hand motion generation and reconstruction baselines on three datasets, consistently outperforming them.

**Strengths:**

The proposed method shines when evaluated against numerous baselines, even in the presence of significant occlusion (Table 1). It is able to handle multimodal conditioning input, making it flexible and able to benefit from various types of known information at inference time.
The proposed method is thoroughly ablated with respect to its components and possible input modalities.
The work includes an honest discussion of its limitations.

**Weaknesses:**

The submission could benefit from more qualitative examples in the supplementary material. This is especially relevant for generative models.

**Questions:**

How does the method differentiate between the left and the right hand in its output?
How does the method perform in the presence of feet, which are often misdetected as hands in egocentric videos?

---

> ### Author Response · Authors · 2025-11-20
> **Response to Reviewer 76ft**
>
> We sincerely thank the reviewer for the very positive assessment. We also appreciate the constructive suggestions. Below, we address the specific questions.
>
> ---
>
> **1. More qualitative results.**
>
> We appreciate this suggestion and have incorporated additional qualitative visualizations in **Appendix C**.
>
> **2. Consistency between left and right hands.**
>
> As described in Line 845 of the paper, “To simplify learning, we horizontally flip input images and corresponding annotations whenever the targeted hand is left, resulting in a right-hand-only network.” Thus, the model always predicts right-hand MANO parameters, which is also a standard practice adopted by prior methods such as HaMeR. For left-hand inputs, we invert the flipping transformation on the predicted right-hand MANO parameters to obtain the corresponding left-hand result.
>
> When more than one hand appears, UniHand relies on the initialization hand pose token within the hand perceptron module to associate each detected hand with its corresponding motion branch, allowing correct binding to the intended hand during generation. We also present a left-hand motion example in **Figure 5**, demonstrating that the model maintains consistent predictions across both hands.
>
> **3. Handling foot mis-detections.**
>
> Thank you for pointing this out. Mis-detections in feet typically arise in 2D keypoint detectors. When 2D keypoints contain such errors (e.g., a foot incorrectly labeled as a hand), the other modalities, such as full-frame vision features and 3D keypoints, provide strong corrective signals. Because the diffusion model jointly generates motion consistent with all conditions, implausible foot motions are naturally suppressed. Furthermore, UniHand does not rely on cropped hand regions, and the hand perceptron processes global visual context, including limb spatial positions, making it easier to disambiguate hands from feet.
>
> Since current hand-pose datasets do not contain foot labeling, we add a qualitative example in **Figure 4** to illustrate UniHand’s robustness in such cases.
>
> ---
>
> Thank you again for the positive assessment and valuable suggestions. We have revised the paper accordingly and included the additional visualizations in the updated PDF.

---

> ### Comment · Reviewer_LJ9z · 2025-11-24
>
> Hi Reviewer 76ft,
> Can you please provide your thoughts on the paper weakness I pointed out?
> Thanks!

---

### Author Response · Authors · 2025-12-01
**Summary of Discussion**

Dear Area Chair,

We appreciate your time and the additional effort required in light of this incident.
In view of the reviews and scores being reverted, we provide a concise summary of all reviewer concerns, how each was addressed during the rebuttal period, and how reviewer scores evolved before the leak on Nov 27, 2025.
Below, we summarize **the timeline of the discussion with each reviewer**. The changes made during the discussion phase are highlighted in blue in the revised paper.

---

### **A. Reviewer 76ft**

**1. Initial Official Review**

`Rating: 8, Confidence: 4`

The main suggestions were to 1) include more qualitative results, along with questions about 2) how the model distinguishes left from right hands and 3) whether it may mis-detect feet as hands.

**2. Our Response (*Nov 20, 17:20 UTC*)**

We 1) added additional qualitative examples in the appendix, 2) clarified the only-right-hand mechanism and how UniHand processes left-hand, 3) explained how multimodal consistency and the hand perceptron module mitigate foot mis-detections, and updated the paper accordingly.

---

### **B. Reviewer LJ9z**

**1. Initial Official Review**

`Rating: 2, Confidence: 4`

The main concern was that our “unified” claim implicitly suggested coverage of generation tasks like HOI grasp generation, while no such results are provided, creating a mismatch between what is claimed in the motivation and what is actually evaluated in the experiments.

**2. Reviewer Clarification (*Nov 13, 22:12 UTC*)**

The reviewer explained that the `Rating: 2` came entirely from a mismatch between the claims and the experiments, not from technical issues, stating that

> “the reason I rate as 2 is simply because there is such a big hole between what is claimed and what is experimented.”
>

At the same time, the reviewer noted that “the method is actually well motivated and experiments look sensible” within the hand pose estimation setting.

**3. Our First Response (*Nov 20, 17:23 UTC*)**

We respectfully **disagree** with the statement that *“there is such a big hole between what is claimed and what is experimented”* and that “*the generation ability of the framework is not tested/reported”*. We also provided a concise clarification of (A) what our “unification” claim means, (B) what types of generation are in the scope of our unification, and (C) how our experiments evaluate the generative capabilities.

**4. Reviewer Reaction (*Nov 24, 23:27 UTC*)**

The reviewer acknowledged the effectiveness of the Joint VAE and conditional denoiser and updated the score to `Rating: 4, Confidence: 4`, but still requested a clearer justification of the motivation.

**5. Our Second Response (*Nov 25, 13:39 UTC*)**

We revised the Abstract and Introduction to avoid references to HOI grasp generation or object-driven synthesis and to avoid confusion about the scope of conditional hand motion synthesis, rewriting the relevant “generation approaches” sentences to focus on multi-modal structured inputs such as 2D/3D skeletons and coarse or incomplete conditions, and adding citations for the corresponding generative prior works.

**6. Reviewer Reaction (*Nov 25, 14:39 UTC*)**

The reviewer commented that

> “the clarification on the generation approach addressed my concerns of the motivation,”
>

and updated the score to `Rating: 6, Confidence: 4`.

**Thus, before the score reversion, this reviewer’s rating increased from 2 to 4 and then to 6 as their main concerns were fully resolved.**

---

### **C. Reviewer 4cS2**

**1. Initial Official Review**

`Rating: 6, Confidence: 3`

The main concerns were the 1) efficiency and computational cost, as well as 2) the need for clearer analysis of how multimodal alignment contributes to performance.

**2. Our Response (*Nov 20, 17:26  UTC*)**

We 1) added detailed training and inference cost comparisons with HaMeR and Dyn-HaMR, clarified the efficiency advantages of UniHand despite the diffusion steps, and 2) introduced an additional ablation study of multimodal alignment.

**3. Reviewer Reaction (*Nov 26, 05:01 UTC*)**

The reviewer stated that

> “authors generally addressed my concerns… I will keep the rating!”
>

And the final score remained `Rating: 6, Confidence: 3`.

---

All issues raised by reviewers had been addressed in detail, and all reviewer-author discussions occurred **before Nov 27, 2025**. Reviewer scores at that time had converged to:

Reviewer 76ft: `Rating: 8, Confidence: 4`

Reviewer LJ9z: `Rating: 6, Confidence: 4`

Reviewer 4cS2: `Rating: 6, Confidence: 3`

We confirm that we strictly followed the ICLR Code of Conduct, did not engage in any prohibited activity, and had no knowledge of the leak before the public announcement.

Thank you again for your time and careful consideration.

With best regards,

The authors of Submission1286

---

### Meta-Review · Area_Chair_j56Y · 2026-01-07

**Summary:**

This paper proposes UniHand, a diffusion-based framework that formulates 4D hand motion estimation and generation as conditional motion synthesis under heterogeneous inputs.

Reviewers found the method technically sound, well motivated, and supported by strong empirical results, particularly under occlusion and temporally incomplete conditions.

The primary concern during review was the clarity and scope of the paper’s “unified generation” claim. Additional questions were raised regarding efficiency analysis and qualitative evaluation. These issues were discussed in detail during the rebuttal phase.

Overall, the reviewers’ concerns were largely addressed, and opinions converged toward a positive assessment. I recommend acceptance as a poster.

**Reviewer Concerns:**

The main concern regarding a potential mismatch between claims and experimental evaluation was addressed through clarification and revisions that better scoped the notion of generation within the paper. The authors also added efficiency analysis, ablation studies, and additional qualitative results, which resolved the remaining technical questions. No substantial unresolved issues remain.

**Reviewer Scores:**

One reviewer [LJ9z] initially rated the paper low due to concerns about claim scope, but subsequently raised the score after clarification. The other reviewers [76ft, 4cS2] maintained positive evaluations throughout.

Overall, reviewer scores converged to the accept range.

---

### Decision · Program_Chairs · 2026-01-26

Accept (Poster)